# The Role of Antioxidants on Wound Healing: A Review of the Current Evidence

**DOI:** 10.3390/jcm10163558

**Published:** 2021-08-13

**Authors:** Inés María Comino-Sanz, María Dolores López-Franco, Begoña Castro, Pedro Luis Pancorbo-Hidalgo

**Affiliations:** 1Department of Nursing, Faculty of Health Sciences, University of Jaén, 23071 Jaén, Spain; mlfranco@ujaen.es (M.D.L.-F.); pancorbo@ujaen.es (P.L.P.-H.); 2Histocell S.L., Bizkaia Science and Technology Park, 48160 Derio, Spain; bcastro@histocell.com

**Keywords:** wound healing, oxidative stress, antioxidant dressing, reactive oxygen species

## Abstract

(1) Background: Reactive oxygen species (ROS) play a crucial role in the preparation of the normal wound healing response. Therefore, a correct balance between low or high levels of ROS is essential. Antioxidant dressings that regulate this balance are a target for new therapies. The purpose of this review is to identify the compounds with antioxidant properties that have been tested for wound healing and to summarize the available evidence on their effects. (2) Methods: A literature search was conducted and included any study that evaluated the effects or mechanisms of antioxidants in the healing process (in vitro, animal models or human studies). (3) Results: Seven compounds with antioxidant activity were identified (Curcumin, *N*-acetyl cysteine, Chitosan, Gallic Acid, Edaravone, Crocin, Safranal and Quercetin) and 46 studies reporting the effects on the healing process of these antioxidants compounds were included. (4) Conclusions: this review offers a map of the research on some of the antioxidant compounds with potential for use as wound therapies and basic research on redox balance and oxidative stress in the healing process. Curcumin, NAC, quercetin and chitosan are the antioxidant compounds that shown some initial evidence of efficacy, but more research in human is needed.

## 1. Introduction

Skin is the largest organ of the human body, forming the outer covering and acting as the first barrier to the external environment against dehydration, chemical or radiological damage and microorganisms’ invasion [1]. A wound is any disruption of the skin layers that alters its structure and function [2]. Immediately after a wound occurs, the process of healing starts. Wound repair is a complex, but dynamic and orderly process, characterized by a series of overlapping phases that interact: (1) coagulation, (2) immune response and inflammation, (3) proliferation and (4) remodeling [3,4]. When this highly regulated process is disrupted, the healing stops, resulting in a chronic wound (also known as a hard-to-heal wound or non-healing wound). A chronic wound is defined as a wound that does not heal in the orderly stages of the healing process, does not heal within three months or is 40–50% unhealed after four weeks of appropriate treatment [5]. 

Reactive Oxygen Species (ROS) are small oxygen-derived molecules mainly produced by the respiratory chain in mitochondria; some of them are hydrogen peroxide H_2_O_2_, superoxide anion O^−^_2_ or peroxide O^2−^_2_. They are oxidizing agents and mayor contributors to cell damage [6,7], but also have beneficial roles and, in particular, play a crucial role in the preparation of the normal wound healing response [8]. Therefore, a suitable balance between low or high levels of ROS is essential. Low levels of ROS are beneficial in protecting tissues against infection and stimulating effective wound healing by production of cell surviving signaling [9,10] but, when present in excess, produce oxidative stress leading to cell damage and a pro-inflammatory status [11]. Redox imbalance occurs whether the levels of ROS exceed the capacity of endogenous antioxidants to scavenge them, which dysregulates the healing process [12]. There are no a clear cut-off point for ROS level in tissues, but for the hydrogen peroxide (the most common ROS) the range 100–250 μM is considered for normal wounds [10,13]. In addition, some studies have reported that level of 10 μM of hydrogen peroxide act as chemo-attractant and stimulates the proliferation of fibroblasts and endothelial cells; at 100 μM stimulates angiogenesis via the production of vascular endothelial growth factor; but at 500 μM led to a pro-inflammatory status through the production of macrophage inflammatory protein 1-α [14].

The antioxidants are chemical compounds that can donate their electrons to other molecules, such as ROS, thus preventing them from taking electrons from other biologically important molecules, such as proteins or DNA [15]. Based on the mechanism of action, there are two types of antioxidant compounds: non-enzymatic and enzymatic. Non-enzymatic antioxidants are low molecular weight compounds, such as vitamin E, vitamin C, glutathione and flavonoids. Enzymatic antioxidants include the superoxide dismutase, catalase, glutathione peroxidases and thioredoxin-1 and -2, among others [15,16]. Antioxidants catalyze a complex cascade of reactions to convert ROS into more stable molecules, such as H_2_O and O_2_, so they are denominate as ROS scavengers. 

Regulation of redox balance through the modulation of ROS and antioxidant levels is a target for new therapies. Antioxidant substances that maintain non-toxic ROS levels in the wound tissues could improve healing [15]. Thus, the interest in using antioxidant compounds for wound treatment is growing, and several biomaterials have been developed and tested [17]. However, most of these new compounds are not well known by clinicians, and their properties and true effects on healing remain unclear.

This review aims to identify the compounds with antioxidant capacity that have been tested for wound healing and summarize the available evidence on their effects.

## 2. Methods

A literature search was conducted in several databases (Scopus, PubMed, CUIDEN, CINAHL, Health and Medical, Web of Science, COCHRANE and ClinicalTrials). The search strategy used the terms: ‘wound healing’, ‘dressing’, ‘oxidative stress’, ‘reactive oxygen species’, ‘ROS’ and ‘oxidation-reduction’. The search was restricted to articles published in the last 15 years (from 2005 to 2020) in English or Spanish (Figure 1). Since this review is intended to cover the full spectrum of research on antioxidants compounds for wound healing, we included any study that evaluated the effects or mechanisms of antioxidants in the healing process (in vitro, animal models and human studies). No specific quality appraisal of the studies was performed.

The studies that met the inclusion criteria were classified according to the type of research (in vitro, animal, human observational or human experimental) and grouped based on the compound with antioxidant activity. A narrative synthesis is reported.

## 3. Results

This review identified seven compounds with antioxidant activity that have been tested for their effect on the healing process, mainly in controlling oxidative stress. These compounds are Curcumin, *N*-acetyl Cysteine, Chitosan, Gallic Acid, Edaravone, Crocin, Safranal and Quercetin (Figure 2). Table 1 displays their origin, chemical and biological properties.

### 3.1. Compounds with Antioxidant Activity

#### 3.1.1. Curcumin

Curcumin is a natural polyphenolic molecule extracted from the Curcuma longa rhizome. This compound has anti-inflammatory, antibacterial and antioxidant properties and specifically improves wound healing [18]. Curcumin affects various stages of the healing process: granulation tissue formation, collagen deposition, remodeling of tissues and contraction of wounds [19]. However, curcumin has extremely low water solubility, limiting its bioavailability and representing a major barrier for therapeutic use. Therefore, curcumin requires the development of suitable carriers to deliver the molecule in a sustained way at therapeutic levels to enhance its bioavailability. Some of the carriers used are hydrogel, nanoparticles, micelles, hyaluronic/oleic acid [20,21] and is a well-known biofilm inhibitor [22].

#### 3.1.2. *N*-Acetyl Cysteine

*N*-acetyl cysteine (NAC) is a sulfhydryl compound and a precursor in the formation of glutathione, which has significant antioxidant activity. NAC plays a role in regulating the redox status in tissues, reducing oxidative stress by transforming the ROS produced by macrophages, endothelial cells and fibroblasts [23,24]. 

#### 3.1.3. Chitosan

Chitosan is a linear polysaccharide composed of d-glucosamine and *N*-acetyl-d-glucosamine derived from chitin (present on the exoskeleton of crustaceans). This compound exhibits several important properties, such as biocompatibility and biodegradability [25,26]. Chitosan acts as a hemostatic agent (through the binding to platelet surface), is antibacterial, and acts as a bioadhesive material (as nanofibers), which is a very promising alternative for wound dressings [27,28,29].

#### 3.1.4. Gallic Acid

Gallic acid belongs to a group of natural polyphenol compounds found in almost all plants, including fruits, leaves and wildflowers. It has gained significant attention for its biological effects, such as its antioxidant, anti-inflammatory and analgesic properties [30,31].

#### 3.1.5. Edaravone

Edaravone (3-methyl-1-phenyl-2-pyrazolin-5-one) is a strong free radical scavenger that suppresses the effect of oxidative stress. As an antioxidant, it has been used to treat acute cerebral infarction because this molecule has a positive influence on cerebral blood flow, suppresses delayed neuronal death, improves core neurologic deficits and shows significantly free radical scavenging properties. However, its low stability and solubility have limited its topical applications [32,33]. 

#### 3.1.6. Crocin and Safranal

Crocin and Safranal (and its precursor picrocrocin) are carotenoid compounds present in Crocus sativus L. (saffron crocus). Both crocin and safranal have a significant antioxidant and free radical scavenging activity. Studies have suggested that these compounds might have anti-inflammatory and antitumoral properties [34,35]. 

#### 3.1.7. Quercetin

Quercetin is a flavonoid compound commonly found in vegetables and fruits. It has strong antioxidant and anti-inflammatory properties, which advocates its possible application in wound healing. In addition, quercetin can inhibit both acute and chronic phases of inflammation [36,37]. In this way, quercetin could regulate two factors that delay the healing process: oxidative stress and inflammation [38,39,40].

### 3.2. Effects on the Healing Process of Antioxidant Compounds

This review describes the findings of 46 studies reporting on the effects on the healing process of these antioxidant compounds and on their efficacy when data are available from the original studies. There are scarce data on the safety on humans. There are 12 studies about curcumin alone or in combination with chitosan or NAC; 12 studies about NAC alone or in combination with curcumin or α-tocopherol; 17 studies about chitosan alone or in combination with curcumin, NAC, plan extracts or gallic acid. Some of the studies are included in several categories of antioxidant substances. Furthermore, there are 4 studies about gallic acid, 2 about Edavarone, 3 about crocin and safranal and 7 studies about quercetin, one of them in combination with oleic acid (Table 2). The results of each compound are grouped according to the type of research: (1) in vitro or animal models and (2) studies in humans. 

#### 3.2.1. Curcumin


In Vitro and Animal Model Studies.


We identified five studies that tested the effect of curcumin on healing using different carriers. Merrell et al. [21] investigated in vitro the feasibility and potential of polycaprolactone nanofibers as a delivery vehicle for curcumin for wound healing applications. The curcumin-loaded nanofibers exhibited antioxidant properties, were cytocompatible and had a cytoprotective effect on human fibroblast cells under conditions of oxidative stress. This study demonstrated the ability of wound closure in vivo in a diabetic mouse model. By day 10, mice treated with curcumin-loaded polycaprolactone nanofibers showed almost 80% wound closure, compared with approximately 60% wound closure in mice treated with polycaprolactone nanofibers. In other words, mice treated with curcumin-loaded nanofibers had a significant increase in the rate of wound closure compared with mice treated only with polycaprolactone nanofibers.

Other researchers [41] evaluated the effect of curcumin delivered by a gelatin microspheres hydrogel (1) on skin healing compared with curcumin + hydrogel (2) or only hydrogel (3) in chemical-induced and genetic diabetic rats. On day 10 post wounding, results of the quantitative analysis revealed that both skin and epidermal thicknesses were 685.6 ± 13.3 μm in group 1, which was much higher than group 2 (432.9 ± 35.3 μm) and group 3 (367.6 ± 7.36 μm), respectively. They concluded that curcumin decreased the wound size, promoted cell migration and improved wound healing.

Gong et al. [42] tested in vitro a dressing composed of curcumin-loaded micelles carried by a hydrogel that could convert to a gel at around body temperature, adhere to the tissue and release curcumin over an extended period. In the animal model study, rats were divided into four groups; wounds treated topically with the application of normal saline (NS), blank micelles in hydrogel (MeH), curcumin encapsulated in polymeric micelles (CureM) or hydrogel loaded with CureM (CureMeH). On day 7 and day 14, wound contractions in the CureMeH group were significantly higher than those in other groups (*p* < 0.05). Increased re-epithelialization was observed in the CureMeH group on day 7 compared to the other groups (*p* < 0.05); however, no significant differences were observed on day 3 (*p* = 0.33) and day 14 (*p* = 0.10) compared with the CureM group. In this way, CureMeH-treated animals showed more significant wound closure, and a higher degree of re-epithelialization, well-organized granulation tissue and significant fibroblastic deposition compared with other groups.

Furthermore, a comparative study was conducted to investigate topically applied curcumin’s temporal wound healing potential in streptozotocin-induced acute diabetic rats. Three groups were used (control, gel-treated and curcumin-treated). Curcumin application increased the wound contraction and decreased the expressions of inflammatory cytokines/enzymes, such as tumor necrosis factor-alpha, interleukin (IL)-1beta and matrix metalloproteinase-9. Curcumin also increased the levels of anti-inflammatory cytokine IL-10 and antioxidant enzymes (superoxide dismutase, catalase and glutathione peroxidase). In addition, curcumin-treated wounds showed better granulation tissue, dominated by marked fibroblast proliferation and collagen deposition and wounds were covered by thick regenerated epithelial layers [43]. 

Pandey et al. [44] developed a composite dressing with antioxidant activity, which could provide a framework for the granulation tissue in full-thickness wounds. This dressing prevents microbial infiltration, keeps moisture and gaseous exchange, and provides a high surface area for cell proliferation. It is composed of polyvinylpyrrolidone nanofibers, cerium nitrate hexahydrate and curcumin*. An in vivo study for open-wound healing was performed in model rats which were randomly divided in four groups of treatment: Group I (gauze), group II (Ciprofloxacin cream), group (polyvinylpyrrolidone nanofibers with cerium nitrate hexahydrate) and group IV (describe previously*). On day 20, only group IV achieved 100% wound healing, whereas groups I, II, III healed to 79%, 90% and 95%, respectively. Furthermore, the dressing of group IV displayed significantly higher (*p* < 0.05) free radical scavenging compared to group III due to synergistic antioxidant activity.

#### 3.2.2. Curcumin and Chitosan 


In Vitro and Animal Model Studies.


Three studies evaluated the effects of curcumin plus chitosan on healing.

In the study by Zhao et al. [19], a new dressing formulated with curcumin, chitosan-alginate and β-cyclodextrin was able to facilitate cutaneous wound healing. Animal experiments were developed with four groups: gauze (G1), chitosan-alginate (G2), chitosan-alginate-curcumin (G3) and chitosan-alginate-cyclodextrin-curcumin (G4). Results showed that G4 (chitosan-curcumin) produced the fastest healing (60.24% ± 5.81% on day 7, 94.39% ± 5.40% on day 14) compared to chitosan-alginate (G2) (53.49% ± 6.74% on day 7, 80.40% ± 9.01% on day 14) and gauze (G1) (40.41% ± 4.64% on day 7, 64.55% ± 7.84% on day 14). However, no significant difference was observed between chitosan-alginate-curcumin (G3) and chitosan-alginate-cyclodextrin-curcumin (G4) (60.72% ± 4.87% on day 7, 92.42% ± 6.24% on day 14). Moreover, in the groups treated with curcumin, wounds exhibited even lower levels of superoxide dismutase and lipid peroxidation. Possible mechanisms of action involved in this accelerated wound-healing process may be attributed to the antioxidative and anti-inflammatory effects of the chitosan-alginate-cyclodextrin-curcumin dressing.

In another study, Abbas et al. [45] tested the synergistic potential of curcumin alone and cross-linked with chitosan-polyvinyl alcohol membranes. The wound healing potential was tested on rabbits that were divided into different groups: untreated (control), treated with 10, 20 and 30 mg of curcumin and its combination with chitosan. When curcumin 30 mg + chitosan-polyvinyl alcohol 80 was applied twice daily, on day 7 the highest percentage of wound reduction was a 52.33% reduction. A similar effect was also recorded on day 14 of treatment, with an 84% reduction. The same effect was observed with curcumin 20 mg + chitosan- polyvinyl alcohol 80. Histological examinations show excellent conditions of skin tissue and their healing. In addition, the results show that as the concentration of curcumin increased, the scavenging property for free radicals also increased, which might also help in the faster recovery of wounds. The maximum scavenging potential of curcumin (53.17%) was recorded when 30 mg per milliliter of crude curcumin was used as an in vitro free radical scavenging against Diphenyl picrylhydrazyl. Along with chitosan-polyvinyl alcohol, scavenging activity was also increased as the concentration increased from 10, 20 to 30 mg curcumin with chitosan-polyvinyl alcohol 80, inhibition potential 72.19%, 83.68% and 87.53%, respectively, was recorded. 

In a study by Mei et al. [46], curcumin was encapsulated in the nanofibers, grafted chitosan and poly (propylene carbonate), by electrospinning, which gradually released the drug. The enhanced wound healing efficacy was confirmed by an in vivo test. The wound closure ratio was up to 85.0 ± 8.2% in the grafted chitosan nanofibers plus curcumin group on day 14, which was only 64.8 ± 12.5% in the control group. Moreover, around 100% (98.7 ± 10.3%) wound closure was observed in this group on day 21, which was 78.3 ± 5.5% in the control group. The difference observed between both groups was significant. Furthermore, in the aforementioned curcumin group, higher granulation scores and higher collagen contents were observed. These results demonstrated that the combination of grafted chitosan and curcumin improved the wound healing process and showed excellent free radical scavenging capabilities.

#### 3.2.3. Curcumin and *N*-Acetyl Cysteine

We identified five studies that tested the effect of curcumin and *N*-acetyl cysteine on healing. 


In Vitro and Animal Model Studies.


Castro et al. [47] developed a novel antioxidant dressing for moist wound care. This product includes an absorbent matrix composed of a galactomannan from a plant-based origin that provides a porous structure and an antioxidant hydration solution composed of curcumin and *N*-acetyl cysteine adequate moisture to the wound bed. It was tested in vitro and in animal wound healing models (pigs). An in vitro oxidative stress injury model was used based on the exposure of human fibroblasts to H2O2. It showed that antioxidant solution exerts a marked protective effect, reducing ROS levels and regulating the expression of inflammation-related genes. Furthermore, the components of this wound dressing were not cytotoxic and demonstrated good biocompatibility. Macroscopic analyses of the in vivo studies in pigs showed a significant progressive reduction in wound area for antioxidant dressing-treated wounds compared to control wounds (14.8 to 45.1% reduction at days 3 and 16, respectively). The antioxidant dressing modulates the inflammatory phase of wound healing, controlling the excessive cell activation and allowing a more orderly transition between the inflammatory, proliferative and remodeling phases of wound healing [47].


Human Studies.


Furthermore, Castro et al. [48] conducted a multicenter, prospective case study series, which reported the results of an antioxidant dressing (curcumin + NAC) on humans in acute and chronic wounds. Overall, 31 wounds were treated with a dressing change every three days. During the 8-week follow-up, nine wounds (29%) completely healed, of which seven (77.8%) were acute and two (22.2%) chronic. The incidence of healing was 77.8% in acute wounds (seven out of nine acute wounds healed) and 9.1% in chronic wounds (two of 22 chronic wounds healed), resulting in an RR of 8.56 of healing for acute versus chronic wounds [95% confidence interval (CI) 2.18–33.56]. This RR means that acute wounds have 8.56 times higher likelihood of healing than chronic wounds. The remaining 22 wounds showed a significant improvement after treatment with the antioxidant dressing. No adverse effect related with the antioxidant treatment was reported. The results obtained in this case study series suggest that the dressing works well for both wounds and that it can be applied to wounds independently of their level of recurrence or severity, effectively eliminating the biofilm and facilitating the progression of the wound out of the inflammatory phase. These findings show that the antioxidant dressing could represent a new and advanced alternative in the dressing landscape [48,49].

Recently, another case series with 31 patients corroborated these results, pointing out the important role of this treatment in the first inflammatory phase of wound healing. The percentage of wound healing rate significantly increased over time (*p* < 0.0001), and was 40%, 63% and 71% at 4, 8 and 12 weeks, respectively. Over the 12 weeks’ follow-up, 16/31 wounds completely healed (50%). Similarly, there was a reduction of pain of 43.8%, 66% and 77% from baseline at 4, 8 and 12 weeks, respectively. No specific adverse effect was reported. Additionally, there was total pain relief in 77% of the wounds by the end of the study. Furthermore, biofilm elimination was achieved in 90% of total wounds by 12 weeks of follow-up, as assessed by clinical observation [50].

#### 3.2.4. *N*-Acetyl Cysteine and α-Tocopherol

Two studies evaluated the effect of *N*-Acetyl cysteine and α-tocopherol in wound healing. 


In Vitro and Animal Model Studies.


Dhall et al. [51] hypothesized that oxidative stress in wounds is a critical component for generating chronicity. They used a diabetic mouse model of impaired healing and inhibited, at the time of injury, two major antioxidant enzymes—catalase and glutathione peroxidase—creating high oxidative stress in the wounds. To reverse chronicity, they treated the wounds with the antioxidants α-tocopherol and *N*-acetyl cysteine and found that oxidative stress was highly reduced, biofilms had increased sensitivity to antibiotics and granulation tissue was formed with proper collagen deposition and remodeling. They found that healing dramatically improved by 30 days after antioxidant agent treatment compared to the wounds treated with a vehicle that can take up to 100 days to close. 

Li et al. [52] used an in vitro biofilm model with a microbiome from diabetic mouse chronic wounds, formed of Pseudomonas aeruginosa (97%). They found that NAC leads to bacterial cell death when used before biofilm is formed, whereas NAC treatment after the biofilm is established, causes biofilm dismantling accompanied by bacterial cell death. As a mechanism, they proposed that NAC could penetrate the bacterial membrane, increase oxidative stress and halt protein synthesis. NAC creates an environment that disrupts wound microbiome biofilm by interfering with bacterial function and survival and disrupting biofilm extracellular polymeric substance integrity [52].

#### 3.2.5. *N*-Acetyl Cysteine

We identified six studies that tested the effect of *N*-acetyl-L-cysteine on healing using different carriers. 


In Vitro and Animal Model Studies.


A study by Ozkaya et al. [23] evaluated the effects of topical and systemic NAC treatment applied to impaired wounds of diabetic rat models. The animals were divided into four groups: group 1 (control), group 2 (topical NAC), group 3 (systemic NAC) and group 4 (topical + systemic NAC). The results showed that both topical and systemic administration of NAC improved wound healing. On day 14, the wounded areas in groups 2, 3 and 4 were found to be smaller than those in group 1 (*p* < 0.05), with no significant difference among groups 2, 3 and 4 (*p* > 0.05). The mean unhealed wounded area was smallest in group 4. This effect of NAC may be related to its antioxidant properties, as this study showed a reduction in oxidative stress parameters. Further studies are needed to understand other mechanisms of NAC in diabetic wound healing.

The study by Oguz et al. [53] showed that topical application of dexpanthenol cream showed no significant wound reduction compared to *N*-acetyl cysteine. Three groups were formed with 10 rats each. The control group undertook no treatment. The second group received dexpanthenol cream, and the third group was administered 3% NAC cream. The epithelialization and granulation rates between the groups were similar in microscopic evaluations (*p* > 0.05). Less inflammation was revealed in the control group (*p* = 0.027). The angiogenesis rate was more remarkable in the NAC group than in the others (*p* = 0.04). Wound closure rates of dexpanthenol and NAC groups were similar and were significantly higher than the control group’s rates (*p* < 0.001). In multiple comparison analysis, dexpanthenol and NAC groups had similar results in terms of wound healing rates (*p* < 0.05), which were both higher than in the control group (*p* > 0.05). Thus, the efficacy of NAC in wound healing is comparable to dexpanthenol, and both substances can be used to improve wound healing. 

In another study, NAC was used to treat burn wounds in vitro and in vivo to investigate mechanisms of action, finding that NAC-treated wounds had better characteristics on re-epithelialization. NAC was studied at different concentrations (0.1%, 0.5% and 3%) in rat models. Results appear to demonstrate that 3% NAC exhibited a better ability to promote wound healing. The percentage of the regenerated epidermis of the 3% NAC-treated wounds was also significantly higher than other groups (*p* < 0.05). The results demonstrated that NAC can potentially promote wound healing activity and may be a promising drug to accelerate burn wound healing [15].

Aktunc et al. [54] investigated whether NAC induces any favorable effects on cutaneous incisional wound healing in diabetic and nondiabetic mice. The study had four groups: 1 (nondiabetic animals without NAC treatment), 2 (nondiabetic animals given NAC), 3 (diabetic animals without NAC treatment) and 4 (diabetic animals with incisional wounds/given NAC). Treated NAC animals, regardless of the presence or absence of diabetes, displayed greater vascular endothelial growth factor expression around the wounded tissue (group 2: 2.92 ± 0.29 vs. group 4: 2.75 ± 0.45; *p* > 0.05); granulation tissue was more pronounced in the group with NAC (1.00 ± 0.00) than in the group without NAC (group 1 0.92 ± 0.29; group 3 0.92 ± 0.29). Furthermore, mean wound-breaking strength content in groups 1 (96.09 ± 10.88) and 3 (83.91 ± 14.53) was found to be significantly lower than those of groups 2 (119.24 ± 17.80) and 4 (100.50 ± 13.00), respectively. 

Moreover, Hou et al. [55,56] developed a novel structured for wound healing, which was composed of nanofibers of biodegradable polycaprolactone (PCL) or polyamide (PA) polymers, *N*-acetyl cysteine (NAC) and collagen (Col). In vitro studies have demonstrated that the structure has good hygroscopicity, large porosity and great biocompatibility. In vivo studies achieved better repair effects and promoted wound healing compared with the control groups. 

In the study with Polycaprolactone (PCL) [55], the percentage of closure area for the PCL-Col/NAC group increased to 84.34% and 93.59% on the 9th and 12th days, respectively, compared with the PCL and PCL-Col groups (*p* < 0.05), which showed a wound size of 53.49% and 75.84% on the 9th day, 57.44% and 85.05% on the 12th day.

In the case of polyamide (PA) [56], the percentage of wound closure for PA-Col/NAC (69.3% ± 11.47%) was higher than that of PA (37.95%± 10.36%) and PA-Col (54.18%± 12.51%) on day 7. On day 14, all the rats in the PA-Col/NAC group presented almost complete wound healing (86.17% ± 5.46%), while the percentage of wound closure for PA (55.15% ± 12.24%) and PA-Col (73.5% ± 7.46%) were much lower.

As described in this section, the positive effects of NAC on wound healing have been shown in several studies; however, further human studies are needed to assess the clinical benefits and the use of NAC as a standard treatment in wound healing.

#### 3.2.6. Chitosan

Six studies had evaluated the effect of chitosan in wound healing. Four are in vitro and animal model studies, and two are human studies. 


In Vitro and Animal Model Studies.


A series of antibacterial, antioxidant and electroactive nanofiber membranes were developed with poly (ε-caprolactone) and quaternized chitosan-graft-polyaniline polymer solutions. In vitro studies showed that these nanofibrous dressings have electroactivity and mechanical properties similar to soft tissue, free radical scavenging capacity, antibacterial properties and biocompatibility. In vivo experiments had an intervention group with a dressing with 15% chitosan, and two control groups, one without chitosan and the other with polyurethane transparent film. On the 14th day, wounds treated with dressings with chitosan and without chitosan were observed to have closure and had about a 9% lead compared with the polyurethane transparent film group in wound contraction (*p* < 0.05). These quantitative data on the wound area demonstrate that dressings with chitosan had a better wound healing effect than the two control groups. Moreover, after 14 days of surgery, granulation tissue in the chitosan group was approximately 225 mm thicker than that in the group without chitosan (*p* < 0.05), while the granulation tissue of the polyurethane transparent film group was thinner than the other groups. Therefore, this study demonstrated that the wound healing process could be significantly accelerated due to nanofiber dressing exhibiting good collagen deposition, granulation tissue thickness and angiogenesis. Thus, this antioxidant dressing with suitable mechanical properties and good free radical scavenging capacity has great potential in wound healing application [57].

In a study by Zhang et al. [58], a novel antioxidant-loaded hydrogel was constructed, prepared by electrostatic interaction between chitosan, heparin (Hep) and poly (γ-glutamic acid) (PGA-H) and loaded superoxide dismutase (SOD). For the in vivo experiment, the rats were divided into three groups: (1) Composite hydrogels of CS; Hep and γ-PGA, (2) Composite hydrogels of chitosan, Hep and γ-PGA loaded with 800 μL of 2 mg/mL SOD solution; and (3) gauze. On days 14 and 21, the results showed that the wounds of group 1 had a better closure ratio of 92.0% ± 3.7% versus group 3 (85.4% ± 2.4%) group 2 (89.8% ± 2.8%). In addition, the most mature and dense collagen deposition was found in the wound treated with group 1, and the epidermal layer in this group was the thickest (epidermal layer thickness of 21.8 ± 3.2 μm versus 7.7 ± 1.3 μm for group 3 and 9.8 ± 1.9 μm for group 2 on day 21. In this way, the antioxidant’s capability of accelerated wound healing by promoting collagen formation and deposition and epidermal formation was demonstrated. Therefore, it could be a promising candidate for wound healing and provides a new method for developing dressings.

In another study, Liu H et al. [29] explored the capabilities of a catechol-modified chitosan film, which demonstrated that it possesses redox activities. The in vivo study with rats found that the wounds treated with the reduced catechol chitosan film had a statistically higher closure than the other groups (chitosan, and oxidized catechol chitosan film. Authors envision that this approach could be incorporated into dressings for advanced wound management. 

Zhao et al. [59] developed an antibacterial, antioxidant and electroactive injectable hydrogel dressing for cutaneous wound healing by using quaternized chitosan-polyaniline and benzaldehyde groups functionalized poly (ethylene glycol)-co-poly (glycerol sebacate). Researchers developed an in vivo study with three groups: (1) Hydrogel with the mentioned ingredients and polyaniline component, (2) Polyurethane transparent film and (3) Hydrogel with the mentioned components without polyaniline component. Results showed that on days 5, 10 and 15, the wound contraction for group 1 had the advantage over the two control groups 2 and 3 (*p* < 0.05). Furthermore, wounds for group 1 were observed to have closure on the 15th day, compared to only half of group 3. Regarding granulation tissue, group 1 was approximately 200 mm thicker than other groups (*p* < 0.01). In short, this hydrogel exhibited excellent antibacterial activity, electroactivity and free radical scavenging capacity that are beneficial to enhancing the wound healing process. The study has demonstrated their potential to significantly promote the in vivo wound healing process, showing excellent blood-clotting capacity and greatly promoted extracellular matrix synthesis, collagen deposition and granulation tissue thickness. These results indicate that they are excellent candidates as bioactive dressings for cutaneous skin wound healing.


Human Studies.


An open multicenter comparative prospective randomized clinical study was conducted to evaluate the safety and efficacy of a chitosan dressing in facilitating healing in a diverse range of chronic wounds. This study involved 90 patients were treated with the chitosan dressing (experimental group; *n* = 45) or traditional Vaseline gauze (control group; *n* = 45). The results showed that the new chitosan dressing was superior to the control dressing for managing chronic wounds. After 4 weeks of treatment, the wound area reduction was significantly greater in the test group (65.97 ± 4.48%) than in the control group (39.95 ± 4.48%) (*p* < 0.001). The average pain score in the test group was 1.12 ± 0.23 and 2.30 ± 0.23 in the control group (*p* < 0.001). The wound depth was also lower in the test group 0.30 ± 0.48 cm than the control group 0.54 ± 0.86 cm (*p* = 0.025). The level of exudate fell, and the dressing could be removed integrally in both the test and control groups. The mean duration of the test group was 27.31 ± 5.37 days and 27.09 ± 6.44 control group days. It was impossible to determine a statistical significance between the test and control groups in the duration of treatment. This is due to the short duration of the study, which was not long enough to allow a higher proportion of chronic wounds to heal completely. However, during the study, 11 wounds had healed (nine in the test group and two in the control group) before the end of the 30-day follow-up period. In particular, the study highlighted the safety of the new chitosan dressing for clinical applications—this dressing is safe and can be effectively used for managing chronic wounds [60]. 

Halim et al. [61] developed a multicenter randomized controlled trial that included two groups: one, treated with a chitosan derivative film, and two, treated with hydrocolloid dressings. The primary outcome of this study was the percentage of epithelization. There was no significant difference in the mean wound epithelialization percentage between groups (*p* = 0.290). On day 13, the mean wound epithelialization in the chitosan derivative film group was 99.17% (95% CI 97.99–100.36) and 99.84 (95% CI 98.64–101.04) in the hydrocolloid group, *p* = 0.437. Patients using the chitosan derivative film experienced more pain during the removal of the dressing than those in the hydrocolloid group (*p* = 0.007). The chitosan derivative film group showed less exudate (*p* = 0.036) and less odor (*p* = 0.024) than the control group. However, there were no significant differences between groups in terms of adherence, ease of removal, wound drainage, erythema, itchiness, pain and tenderness. This study concluded that the chitosan derivative film was equivalent to hydrocolloid dressing and can be an option in the management of superficial and abrasion wounds.

#### 3.2.7. Chitosan and Plant Extracts

Two studies evaluated the effect of chitosan with plant extracts. 


In Vitro and Animal Model Studies.


Colobatiu L et al. [27] developed a chitosan film formulation as a dressing material for diabetic wounds. This formulation was loaded with bioactive compounds (an extract mixture of Plantago lanceolata, Tagetes patula, Symphytum officinale, Calendula officinalis and Geum urbanum). In vitro studies provided a beneficial moist wound environment, reducing the risk of dehydration and favoring the closure of the wounds. Additionally, its formulation exhibited good antioxidant activity, as well as a proliferative effect and adequate biocompatibility. For in vivo studies with rats, the animals were divided into groups. The first group of animals was treated with iodine povidone, the second group with a blank chitosan film formulation (without bioactive compounds), while the bioactive compounds-loaded chitosan film was applied on the wounds created in the third group of animals. The wounds treated with the chitosan film dressings, and especially with the bioactive compounds-loaded chitosan film formulation, healed in a significantly shorter period compared to the wounds induced in the second group. On day 14, the wounds of the third group were almost completely healed (97.47%), whereas those in the second group showed only a 61.07% healing rate. The results indicated that the chitosan-based dressings, especially the bioactive compounds-loaded dressings, were effective and beneficial in accelerating the healing of diabetic wounds [28].

Rocasalbas et al. [62] used polyphenols from Hamamelis virginiana for cross-linking chitosan/gelatin blends to develop a bioactive hydrogel intended for application as a dressing. The potential of these hydrogels for chronic wound treatment was evaluated in vitro, assessing their antibacterial and inhibitory effect on ROS. Further investigation into this extract would be recommended before justifying in vivo trials. Furthermore, another study investigated polyphenol-rich acetone/water extracts from Hamamelis virginiana in vitro. Polyphenols from this plant were found to act as efficient scavengers of radical and non-radical reactive species, preventing the accumulation in the chronic wound site. Even though no in vivo trials in a wound setting were performed, the results indicate that Hamamelis virginiana merits further investigation [63]. 

Regarding other plant extracts, Bektas N et al. [64] investigated the effects of adding vitexin to a chitosan-based gel for accelerating the wound healing process. The vitexin-containing gel showed significantly improved healing activity in both in vitro and in vivo studies, promoting skin cell proliferation and regeneration. In the in vivo experiments with rats, animal treatment groups were formed as follows: (1) control (no treatment), (2) the positive control (Madecassol treatment), (3) group chitosan treatment and (4) treated group with a chitosan-based gel formulation containing vitexin. There were no statistically significant changes in the evaluation of 21-day groups (*p* = 0.214). Wound healing was detected in all groups. The mean of epidermal-dermal regeneration, granulation and angiogenesis on day 21 was: group 1 (33.83 ± 0.75), group 2 (3.17 ± 1.60), group 3 (3.67 ± 1.03) and group 4 (2.67 ± 1.21). The vitexin formulation was found to provide reepithelization and wound healing in a shorter time. This was attributed to its antioxidant and anti-inflammatory effects, in synergy with the pro-healing activity of chitosan. However, further investigation may be required for clinical use. 

#### 3.2.8. Chitosan and Gallic Acid

We identified four studies that tested the effect of chitosan and gallic acid on healing using different carriers. 


In Vitro and Animal Model Studies.


Thi et al. [31] designed an injectable gelatin hydrogel combining gallic acid with a gelatin polymer backbone. Due to the ROS-scavenging properties, the hydrogels protected the cells from oxidative damage and effectively accelerated the wound healing process with high-quality healed skin. For animal experiments, rats were randomly divided into three groups: (1) control group with Dulbecco’s phosphate-buffered saline, (2) group with hydrogel of gelatin-hydroxyphenyl propionic and (3) hydrogel with gallic acid-conjugated gelatin introduced into gelatin-hydroxyphenyl propionic. 

On day 14, the gel groups 2 and 3 demonstrated a better healing effect than group 1, with the wound area almost mended by gel treatment, while approximately 3% wound closure was left in treatment group 1. The group treated with gallic acid effectively showed improved re-epithelialization and wound remodeling compared to the other groups. Researchers believe that this injectable ROS-scavenging hydrogel has great potential for wound treatment and tissue regeneration, where oxidative damage by ROS contributes to the pathogenesis

In another study, Singh et al. [65] investigated the wound healing activity of the Terminalia bellerica fruit ethanolic extract compared with its active constituent, gallic acid, in experimentally induced diabetic animals. Rats were divided into seven groups: group I normal control (saline 5 mL/Kg), group II normal experimental (Terminalia bellerica extract 400 mg/Kg), group III normal experimental (gallic acid 200 mg/Kg), group IV diabetic control (vehicle only 5 mL/Kg), group V diabetic experimental (Terminalia bellerica extract 400 mg/Kg), group VI diabetic experimental (gallic acid 200 mg/Kg), group VII diabetic standard (Vitamin C 200 mg/Kg).

The administration of fruit extract or gallic acid was ingested orally. The percentage wound contraction on the 5th day post wounding of groups II, III, VI and VII was significantly improved (*p* < 0.05) compared to control groups I and IV. The same pattern was found on the 5th, 10th, 15th and 20th day post wounding. On day 20, wounds in treated groups II, III, V and VI were shown fully developed epithelization, new blood vessel formation and deposition of collagen protein. Results obtained suggested that the role of fruit extract of Terminalia bellerica and its dynamic chemical constituent gallic acid improved wound healing in healthy and diabetic rats, but a higher wound healing effect was reported with gallic acid. Even so, studies should be repeated in models of extended hyperglycemia before drawing conclusions.

Yang et al. [66] conducted an in vitro study of the effect of gallic acid on wound healing in normal and hyperglucidic human keratinocytes and fibroblasts. Results showed that gallic acid has antioxidant properties (by upregulating the expression of antioxidant genes), stimulates cell migration on fibroblasts and activates healing factors, such as focal adhesion kinase or *N*-terminal kinase. They concluded that gallic acid might be a potential wound healing agent to treat wounds resulting from metabolic complications.


In Vitro and Ex Vivo Studies.


In the study by Stefanov et al. [30], multifunctional hydrogels for chronic wound application were produced by enzymatic cross-linking of thiolated chitosan and gallic acid. The hydrogels combine several beneficial wound healing properties, controlling the matrix metalloproteinase and myeloperoxidase activities, oxidative stress and bacterial contamination. In vitro studies revealed greater than 90% antioxidant activity and myeloperoxidase and collagenase inhibition by up to 98 and 23%, respectively. Furthermore, ex vivo studies with venous leg ulcer exudates confirmed the inhibitory capacity of the dressings against myeloperoxidase and matrix metalloproteinase. Herein, these novel biopolymer-dressing materials illustrate an integrated strategy for chronic wound management, upgrading the widely accepted concept for maintaining the moisture environment of the wound with bioactivities to control major factors governing wound chronicity.

#### 3.2.9. Edaravone

Two studies evaluated the effect of edaravone on wound healing.


In Vitro and Animal Model Studies.


This study investigated the combined activity of topical edaravone and alginate with different doses on wound healing in diabetic mice using a hydrogel with potent free radical scavenging ability. The experiment reports that a low dose of hydrogel accelerated wound healing. In contrast, a high dose of edaravone might hamper the healing. On the 10th day post wounding, 96.6% of wounds were repaired with the low-dose treatment. Furthermore, the wound healing rate of the low-dose treatment was 3-fold higher than that of the high-dose on the 5th post wounding day. Those results might be a key factor in the translational application of edaravone in wound healing. Thus, the alginate-based nanocomposite hydrogel is promising for diabetic wound healing because of its innate alginate activity and sustained edaravone release [32].

Naito et al. [67] investigated whether accelerated wound closure occurred in the edaravone group compared with a petroleum jelly group in diabetic mice. On day 7, the rate of wound closure was significantly greater in mice treated with edaravone than in mice treated with petroleum jelly, 68% and 55%, respectively (*p* = 0.0019). Histologically, more abundant blood vessels were observed in the edaravone-treated wound sites than in the control wound sites.

#### 3.2.10. Crocin and Safranal

We identified three studies that tested the effect of crocin and safranal on healing using different carriers. 


In Vitro and Animal Model Studies.


Zeka K et al. [68] tested the performance of a hydrogel enriched with antioxidants compounds (kaempferol and crocin) isolated from saffron crocus petals. The results showed that hydrogels had good biocompatibility with in vitro cultured fibroblasts. It was observed that cells grew faster in antioxidant-enriched hydrogels than in the control; it demonstrated the capacity of the compounds extracted from saffron petals to stimulate fibroblast expansion. The concentrations of crocin and kaempferol glycosides used in the hydrogels were sufficiently high to exert antioxidant activity and protect against the effects of reactive oxygen species. These hydrogels could be considered an easy, cheap, enriched delivery system with applications in treating difficult wounds or wound healing without the need for complicated guidelines. These results could be a preliminary preparation for future in vivo applications. 

Other studies indicated that saffron might be considered a therapeutic option in managing burn wounds because of its beneficial biological effects on tissue regeneration. Khorasani el at. [34] and Alemzadeh et al. [35] performed in vivo studies to evaluate the effect of Crocus sativus extract compared to silver sulfadiazine on the rate of burn wound healing in rats. Saffron significantly enhanced vascularization and improved the proliferation and migration of fibroblasts. Inflammation decreased and increased the rate of wound closure, including re-epithelialization and wound contraction compared to other treated wounds. Furthermore, treatment with saffron decreased the neutrophil counts and reduced free radicals and ROS in the burn environment.

In the study of Khorasani et al. [34], animals were divided into four groups. Group 1 was the control group and no topical agent was applied. Group 2 used a base cream only without an effective agent. Group 3 used 1% silver sulfadiazine, and group 4 used a 20% saffron cream. The average wound size after 25 days was 5.5, 4, 0.9 and 4.1 cm^2^ in groups 1, 2, 3 and 4, respectively. There were significant differences between the saffron group and the other three groups in this aspect (*p* < 0.05). There was no observed statistical difference among the control, base cream and silver sulfadiazine groups in wound size on the 25th day of treatment.

In the study by Alemzadeh et al. [35], the groups were: untreated (no medication), treated topically with 1% silver sulfadiazine and a saffron pomade group. The animals treated with saffron showed more significant wound closure (83.04 ± 1.36% on day 7; 98.78 ± 2.1% on day 14) than the silver sulfadiazine treated (57.57 ± 2.8% on day 7, *p* < 0.001; 98.4 ± 1.2% on day 14) and untreated groups (35.53 ± 3.5%, on day 7, *p* < 0.001; 80.6 ± 1.25% on day 14, *p* < 0.001) on day 7 and day 14, respectively.

In general, saffron significantly enhanced vascularization and improved the proliferation and migration of fibroblasts. Saffron could be considered an optimal option for promoting skin repair and regeneration in burn care due to the antioxidant and anti-inflammatory effects of saffron demonstrated by in vivo and in vitro studies, but further studies are needed to elucidate the exact mechanism of saffron in wound healing.

#### 3.2.11. Quercetin

We identified six studies that tested the effect of quercetin on healing using different carriers. 


In Vitro and Animal Model Studies.


Ajmal et al. [69,70] used electrospinning to develop a dressing composed of poly (e-caprolactone) nanofiber with ciprofloxacin hydrochloride and quercetin. In vitro studies have demonstrated some important features of this nanofiber, such as inhibiting bacterial load and excess free radical activity in the wound area, preserving the functionality of erythrocytes by protecting against lipid peroxidation and promoting fibroblast viability by shielding against oxidative damage. The wound healing efficacy of this nanofiber was assessed using a full-thickness wound model in rats, which were divided into four groups. Group-1 was covered with gauze, group-2 was treated with poly (e-caprolactone), group 3 with poly (e-caprolactone)/Ciprofloxacin hydrochloride and group 4 with poly (e-caprolactone)/Ciprofloxacin hydrochloride Quercetin nanofiber membrane. At the end of 4th day, group 4 displayed significant (*p* < 0.001) healing (43.98%) compared with group 3 (30.56%), which might be due to the effective attenuation of the ROS during the inflammatory phase by burst-release of quercetin. Overall, the results found that quercetin with this nanofiber effectively reduced any possible infection and promoted collagen synthesis by preventing oxidative damage of fibroblasts.

Tran et al. [38] developed a new antimicrobial-antioxidant coating for wound dressing material that contained antimicrobial silver and, as an active ingredient, the antioxidant flavonoid quercetin. In vitro antibacterial assay results showed that the synergism of Ag and quercetin enhanced the antibacterial ability of quercetin, and strong antioxidant activity was also demonstrated. The dressing was applied to surgical wounds created on mice and showed faster wound closure than rats treated only with cotton gauze. On day 12, the wounds with antioxidant coating nearly achieved complete closure (97 ± 2.6%) compared to those in the control group (89.8 ± 2.3%). Thus, it was associated with enhanced tissue remodeling and neo-angiogenesis and a reduction in tissue inflammation. This new coating should be further investigated as a promising material for wound dressing applications

Gomathi et al. [39] found that a quercetin-incorporated collagen matrix healed wounds in rats and scavenges the free radicals more effectively than normal collagen matrix. In addition, animals treated with quercetin showed a better percentage of wound contraction (20 ± 1.77) on day 7 when compared to collagen-treated wounds (39 ± 1.75).

Yin et al. [40] investigated the novel effect of topical application of quercetin in pressure ulcers using an animal model. The study demonstrated that quercetin effectively promoted wound closure and reduced immune cell infiltration and pro-inflammatory cytokine production. On day 14, quercetin-treated wounds were nearly closed, while the wound treated in the control group only recovered 80%. Therefore, results supported that quercetin could be a potential therapeutic agent for pressure ulcers. 

Kant et al. [71] designed a study for the detailed evaluation of the wound healing potential of quercetin at different concentrations. Results showed that 0.3% quercetin promoted better healing compared to that of control and other quercetin doses in rats. This concentration accelerated wound healing by rapid wound contraction. It controlled modulation of inflammatory and anti-inflammatory cytokines, enhanced neovascularization and fibroblast proliferation with marked collagen deposition, rose myofibroblast formation and improved the antioxidant status at the wound site.

#### 3.2.12. Quercetin and Oleic Acid

One study has tested the effect of quercetin and oleic acid in humans. Gallelli et al. [72] evaluated the clinical efficacy and safety of a nano-hydrogel embedded with quercetin and oleic acid for treating lower limb skin wounds in patients with diabetes mellitus. Researchers developed an experimental study with 56 patients randomized to receive treatment with hyaluronic acid (group A) or nano-hydrogel embedded with quercetin and oleic acid (group B). After 1 month of treatment, they documented complete healing in nine patients in group A (32.2%) and in 19 of group B (67.8%) (*p* < 0.01), with a shorter healing time (*p* < 0.01) in group B (10 ± 5 days) compared to group A (25 ± 4 days), depending on the wound size, wound depth and general patient condition. Within 60 days, we documented a total re-epithelization in 16 patients in group A (57.1%) and 26 patients in group B (92.8%) (*p* < 0.01). Furthermore, results described an increase in keratinocyte proliferation after their exposition to quercetin plus oleic acid. Hence, this new topical pharmacological formulation may be effective and safe for skin repair in patients with diabetes. Nonetheless, other clinical trials should be performed to validate these data in a large group of patients.

## 4. Discussion

This review identified seven compounds with antioxidant properties for wound healing (curcumin, *N*-acetyl cysteine, chitosan, gallic acid, edaravone, crocin and safranal and quercetin). The effect of these substances on the healing have been tested with different experimental designs (in vitro, animal models or human clinical studies). Most of them were in vitro or animal model studies; but for curcumin plus NAC, chitosan and quercetin there are also human studies published or ongoing. 

Based on how the antioxidant compound is delivered to the wound tissues, we have identified two technologies used in these studies: hydrogel and nanofibers. Hydrogel were frequently used in wounds since a number of years, as a moisture environment dressing, but there is now a new interest in their use as carrier of bioactive products. Curcumin, curcumin + NAC, chitosan, chitosan + gallic acid and edaravone have been tested using hydrogels. 

Nanofibers are a new emerging technology of great interest for wound therapy. This technology allows the active compound (antioxidant) to be encapsulated in an inert material (mostly polycaprolactone) that can be converted in a fiber by electrospinning. These nanofibers can gradually release the active compound once applied over the wound. There are studies using nanofibers as carrier for curcumin, curcumin + chitosan, NAC and quercetin.

Most of the available evidence was generated in animal studies. Three species of animals have been used to produce wound model for testing antioxidant compounds: rats, mice (diabetic and non-diabetic) and rabbits. In this review, we have identified several outcomes measured to evaluate the efficacy of the antioxidant tested; these outcomes include percentage of wound completely healed, reduction in wound area or size, area of re-epithelization, granulation scores, granulation rates, angiogenesis and collagen formation. All but the first (complete healing) were surrogate outcomes to evaluate the efficacy of the therapy in healing the wounds. Overall, these studies in animal models concluded that the antioxidant treated groups had an improvement in most of the outcomes measured that could be associated with the reduction of the oxidative stress. This agrees with the findings in a recent review on the effect of polyphenols (a type of antioxidant) on chronic wounds, reporting that the reduction in ROS promote the wound healing by the activation of pro-healing and anti-inflammatory genes pathways [12]. 

Similarly, for human studies the outcomes reported in the studies were: complete healing (percentage of epithelization), wound area, wound pain, wound depth and level of exudate. From the clinician’s perspective, all of these outcomes may be relevant in wound management, although complete healing is the most important and perhaps the only relevant from patient’s perspective. 

There is evidence in humans for curcumin + NAC, chitosan and quercetin. For curcumin + NAC, two observational studies in patients with several type of wounds [48,50] have shown an increase in the percentage of complete healing and a reduction in wound size in patients treated with this combination of antioxidants; so this is one of the most promising therapies. Currently, a RCT is ongoing with the aim to compare the effect on the healing of chronic wounds of an antioxidant dressing with curcumin + NAC with the standard dressing treatment [73]. 

For quercetin, a small RCT tested a hydrogel with quercetin plus oleic acid versus a hyaluronic acid dressing in diabetic patients with lower limbs ulcers [72]. This study found a significant increase in complete healed wounds at two months that can be attributed to the antioxidant treatment. Nevertheless, this finding need to be confirmed with large RCT and in patients with other type of wounds. 

For chitosan, there is evidence from experimental studies. A RCT found that chitosan dressing had better outcomes in wound healing compared to traditional vaseline gauze in wound size, pain, wound depth and exudate, but it is inconclusive in complete healing [60]. However, the value of this evidence is reduced because of the use of gauze (a non-recommended treatment) in the control group. Another RCT compared chitosan with hydrocolloid dressing in chronic wounds [61]. This study found that chitosan had high percentage of re-epithelization, but not better than those treated with hydrocolloids; therefore, it is unclear whether the antioxidant effect of chitosan have an impact on healing. Again, these results are in line with the conclusions of a systematic review on the effect of chitosan, which concluded that chitosan may perform better that traditional gauze, but not better than wet dressings [74]. This uncertainty could be solved in the future, because three RCT testing chitosan are ongoing. One in patients with diabetic foot ulcers (ClinicalTrials.gov Identifier: NCT04178525); the second, test the combination of isosorbide dinitrate spray and chitosan in diabetic foot ulcers (ClinicalTrials.gov Identifier: NCT02789033); and a third to evaluate the efficacy and safety of sericin and chitosan cream for preventing and limiting the progression of pressure ulcers (ClinicalTrials.gov Identifier: NCT04559165). 

We highlight that current research on new biomaterials for wound therapy point to two innovations with great potential for future clinical use: first, the use of nanofibers as a versatile way to deliver active compounds to wound tissues; second, the major role of the redox balance in the normal healing process; chronic or hard-to-heal wounds were often stagnate in the inflammatory phase due to of oxidative stress [12]. In this context, therapies based on the topical use of antioxidants compounds might regulate the redox balance, decreasing the inflammatory status, allowing the wound to continue the proliferative phase of healing [12,17,75]. From this point of view, these therapies should be used not for every wound, but as a second-line therapy for hard-to-heal wounds. The current evidence mainly from in vitro and animal model research warrants the development of clinical studies in humans with experimental design to test the efficacy and the safety of antioxidants.

This review has some limitations to consider. Since this is an emerging issue, several types of studies have been included, not only human clinical trials, but also in vitro and animal studies, which has led to a great heterogeneity in the results and methods used, making synthesis difficult. None of the studies measured directly the levels of ROS in tissues; instead, most of them measured oxidative stress parameters (lipid peroxidation or SOD) and some clinical parameter of healing. Monitoring the concentration of ROS in wound tissues and its changes with antioxidants therapies might be a topic to consider for future research. 

Safety and adverse effects of these therapies are poorly considered. None of the animal studies were designed to record possible adverse effects, and only a few of the human studies address them. Therefore, this is another important issue for future clinical research in human, in order to obtain knowledge about potential side effects of decreasing ROS level in wounds. 

As a conclusion, this review offers a map of the research on some of the antioxidant compounds with potential for use as wound therapies and basic research on redox balance and oxidative stress in the healing process. Curcumin, NAC, quercetin and chitosan are the antioxidant compounds that shown some initial evidence of efficacy, but more research in humans is needed. 

## Figures and Tables

**Figure 1 jcm-10-03558-f001:**
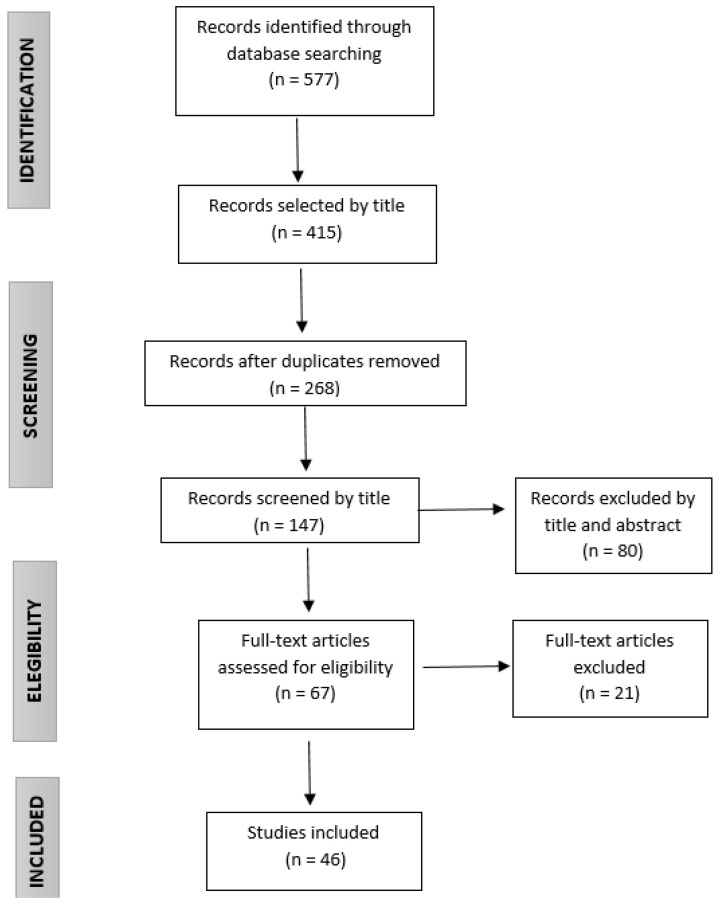
Flow chart of the selection of reviewed articles.

**Figure 2 jcm-10-03558-f002:**
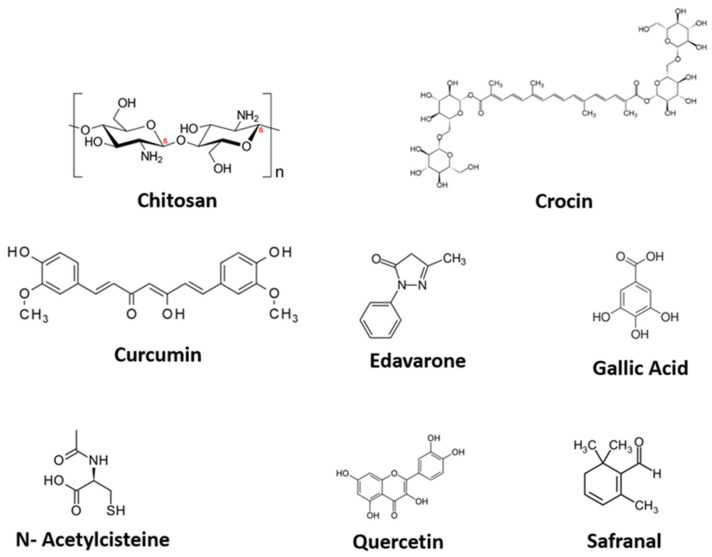
Chemical formula of the antioxidant substances.

**Table 1 jcm-10-03558-t001:** Compounds with antioxidant activity.

Compound	Type	Chemical Properties	Biological Properties	Origin
Curcumin	Polyphenolic	Non-water-soluble	Anti-inflammatory, antibacterial and antioxidant	Vegetal Curcuma longa rhizome
Chitosan	Polysaccharide (derived from chitin)	Water-soluble	Highly biocompatible Hemostatic, antibacterial and antioxidant	Animal Exoskeleton from crustaceans
*N*-acetyl Cysteine (NAC)	Sulfhydryl	Water-soluble	Precursor in the formation of glutathione (GSH), antioxidant	Modified form of the amino acid l-cysteine
Gallic Acid	Polyphenol	Soluble in alcohol, ether, acetone	Antioxidant, anti-inflammatory, analgesic	Vegetables, fruits, leaves and wildflowers.
Edaravone	3-methyl-1-phenyl-2-pyrazolin-5-one	Soluble in hot water and hot alcohol	Free radical scavenger, antioxidant	Chemical synthesis
Crocin and Safranal	Carotenoid	Lipophilic (poor water solubility)	Antioxidant, anti-inflammatory, antitumoral	Vegetal of saffron crocus (Crocus Sativus)
Quercetin	Flavonoid (polyphenol)	Non-water-soluble	Antioxidant, anti-inflammatory	Vegetables and fruits

**Table 2 jcm-10-03558-t002:** Studies about the effect of the antioxidant compounds on the wound healing process.

Compound	Author	Year	Carrier	Type of Study		
				In Vitro	Animal Model	Human
Curcumin	Merrell [21]	2009	Polycaprolactone nanofibers	X	X	
	Liu [41]	2018	Gelatin microspheres	X	X	
	Gong [42]	2013	Hydrogel	X	X	
	Kant [43]	2014			X	
	Pandey [44]	2020	Polyvinyl pyrrolidone nanofibers	X	X	
Curcumin + Chitosan	Zhao [19]	2019	β-cyclodextrin	X	X	
	Abbas [45]	2019			X	
	Mei [46]	2017	Nanofibers polypropylene	X	X	
Curcumin + *N*-acetyl-cysteine	Castro [47]	2015	Galactomannan	X	X	
	Castro [48]	2017	Galactomannan			X (Obs)
	Jimenez-Garcia [49]	2018	Galactomannan			X (Obs)
	Jimenez-Garcia [50]	2021	Galactomannan			X (Obs)
*N*-acetyl cysteine + α-tocopherol	Dhall [51]	2014		X	X	
	Li [52]	2020	Microbiota	X		
*N*-acetyl cysteine	Ozkaya [23]	2019			X	
	Oguz [53]	2015			X	
	Tsai [24]	2014		X	X	
	Aktunc [54]	2010		X	X	
	Hou [55]	2019	polycaprolactone	X	X	
	Hou [56]	2020	polyamide	X	X	
Chitosan	Jiahui [57]	2020	polycaprolactone	X	X	
	Zhang [58]	2018	hydrogel poly (γ-glutamic acid) and heparin	X	X	
	Liu [29]	2018	catechol-modified chitosan film	X	X	
	Zhao [59]	2017	Hydrogel (polyaniline)	X	X	
	Mo [60]	2015				X (Exp)
	Halim [61]	2018				X (Exp)
Chitosan + plan extracts	Colobatiu [27,28]	2019	Polymer and polyvinyl alcohol	X	X	
	Rocasalbas [62]	2013	Gelatin hydrogel	X		
	Díaz-González [63]	2012		X		
	Bektas N [64]	2020	Hydrogel	X	X	
Chitosan + gallic acid	Thi [31]	2020	Gelatin hydrogel	X	X	
	Singh [65]	2019			X	
	Yang [66]	2016		X		
	Stefanov [30]	2016		X		X (ExV)
Edaravone	Fan [32]	2019	Alginate hydrogel	X	X	
	Naito [67]	2014			X	
Crocin and Safranal	Zeka [68]	2017	hydrogel	X		
	Khorasani [34]	2008			X	
	Alemzadeh [35]	2018			X	
Quercetin	Ajmal [69]	2019	poly (e-caprolactone)	X		
	Ajmal [70]	2019	poly (e-caprolactone)		X	
	Tran [38]	2019			X	
	Gomathi [39]	2002			X	
	Yin [40]	2018			X	
	Kant [71]	2020			X	
Quercetin + Oleic acid	Gallelli [72]	2020				X (Exp)

X: Type of study; Exp: Experimental; ExV: Ex Vivo; Obs: Observational.

## Data Availability

Data available on request due to restrictions.

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
