# Peer review of "The Role of Antioxidants on Wound Healing: A Review of the Current Evidence"

_jcm, 2021, doi:10.3390/jcm10163558_

Round 1
Reviewer 1 Report
1. The authors aimed to review "THE ROLE OF ANTIOXIDANTS ON WOUND HEALING", it would be important to discuss the mechanisms and pathways by which those antioxidants work.
2. As stated, high levels of ROS could cause cell damage but low levels of ROS are beneficial in protecting tissues against infection and stimulating effective wound healing in normal wound healing response. A suitable balance between low or high levels of ROS is essential.
So what is the threshold or index for high or low ROS level? Was the level of ROS monitored in all studies cited in the manuscript?
3. It would be good to know the changes of ROS level after treatment with those antioxidants, and how the changes correlated with the wound healing?
4. Any potential adverse effects if the level of ROS is decreased to extremely low level or abolished by antioxidants during wound healing? For example, Curcumin application increased the wound contraction...as described (line 182).
Author Response
Dear reviewer,
Thank you very much for taking the time to review the article and making comments and suggestions. _________________________________________________________________________
Point 1: The authors aimed to review "THE ROLE OF ANTIOXIDANTS ON WOUND HEALING", it would be important to discuss the mechanisms and pathways by which those antioxidants work.
Response: In the introduction section, we have added a paragraph explain the basic mechanisms and types of antioxidants function, with some references.
Point 2: As stated, high levels of ROS could cause cell damage but low levels of ROS are beneficial in protecting tissues against infection and stimulating effective wound healing in normal wound healing response. A suitable balance between low or high levels of ROS is essential.
Response: This is right. We have added some specific information about the level of ROS that can be considered as suitable for wound healing, according current research. More information about the balance in ROS is also included.
So what is the threshold or index for high or low ROS level?
Response: We have added specific information about the level of ROS that could be considered as low in wound; with a range from 100 to 250 mcM.
Was the level of ROS monitored in all studies cited in the manuscript?
Response: The levels of ROS were not measured in the studies reported in this review. We have explained this as a limitation, in the Discussion section.
Point 3: It would be good to know the changes of ROS level after treatment with those antioxidants, and how the changes correlated with the wound healing?
Response: Direct measurement of ROS in tissues is extremely difficult because of their high reactivity and short lifetimes; for this reason none of the studies in human or animal reported data of ROS levels, but for other parameters (oxidative stress) or for wound healing (rate of healing, time). Perhaps monitoring the level of ROS in wounds could be issue for research in the future. We have mentioned this in the discussion.
Point 4: Any potential adverse effects if the level of ROS is decreased to extremely low level or abolished by antioxidants during wound healing? For example, Curcumin application increased the wound contraction...as described (line 182).
Response: Extremely low level of ROS could impair the healing process, which would be adverse effects of the treatment with antioxidants. However, from a theoretical point of view it is highly unlikely that topical antioxidants may produce such a reduction in ROS levels. None of the studies included in this review reported adverse effects on wounds.

Reviewer 2 Report
In this systematic review, the authors tried to identify and summarize the compounds with antioxidant properties that have been tested for wound healing and the available evidence on their effects. It is an interesting topic and will be benefit the clinical application in the future. It is suggested to improve the following points.
- Item in Table 1, ‘Chemical Properties’ for Gallic Acid, Edaravone, Crocin and Safranal are insufficient.
- In 3.2 section, it should not be ended in listing the results in vitro and animal model studies as well as human studies. If available, it is preferred to compare the safety, efficiency as well as the advantage and limitation for the different compounds and their combinations?
- In discussion section, it should not be repeated so much which mentioned in former section even if the important clinical trials. The clear and comprehensive discussion with a logic flow will be benefit much for the readers. So, it is suggested to re-arrange and simplify this section with such as the brief follow-up of clinical trials, the advantage and limitation, which could point out the future research directions.
Author Response
Dear reviewer,
Thank you very much for taking the time to review the article and making comments and suggestions. _________________________________________________________________________
In this systematic review, the authors tried to identify and summarize the compounds with antioxidant properties that have been tested for wound healing and the available evidence on their effects. It is an interesting topic and will be benefit the clinical application in the future. It is suggested to improve the following points.
Point 1: Item in Table 1, ‘Chemical Properties’ for Gallic Acid, Edaravone, Crocin and Safranal are insufficient.
Response: It is true that the table was incomplete. We have completed the table 1 with the chemical properties for these compounds (Gallic Acid, Edaravone, Crocin and Safranal).
Point 2: In 3.2 section, it should not be ended in listing the results in vitro and animal model studies as well as human studies. If available, it is preferred to compare the safety, efficiency as well as the advantage and limitation for the different compounds and their combinations?
Response: We agree that organizing the results in this way (efficacy, safety and advantages) should be the best option. However, because the type of data provided by the original studies included in this review this way is not possible. Most of the studies are in vitro and other report heterogeneous results on animal models. In this section, we have tried to summarize the main findings over the healing process. This is a limitation of the review, which we consider in the discussion.
Point 3: In discussion section, it should not be repeated so much which mentioned in former section even if the important clinical trials. The clear and comprehensive discussion with a logic flow will be benefit much for the readers. So, it is suggested to re-arrange and simplify this section with such as the brief follow-up of clinical trials, the advantage and limitation, which could point out the future research directions
Response: We have completely revised and modified the structure of the Discussion according reviewers’ suggestions. Now, this section explains clearly, which are the main findings of this review, the clinical implications and the limitation of the research.

Reviewer 3 Report
The ms covers the topic of use of anti-oxidants compounds for wound healing.
I suggest to modify the discussion section in order to put more in emphasis the possible application in wound therapy.
I also suggest to check the english.
Author Response
Dear reviewer,
Thank you very much for taking the time to review the article and making comments and suggestions. _________________________________________________________________________
The ms covers the topic of use of antioxidants compounds for wound healing.
Point 1: I suggest to modify the discussion section in order to put more in emphasis the possible application in wound therapy.
Response: We have completely revised and modified the discussion, according to both reviewers’ suggestions. Now, we think that this section explain clearly the main findings of this review and the limitations.
Point 2: I also suggest to check the english.
Response: We have checked the English and corrected some mistakes identified.

Round 2
Reviewer 1 Report
Thanks for making the changes. The manuscript is now a lot more readable.